# OpenReview forum: "Utility Engineering: Analyzing and Controlling Emergent Value Systems in AIs"
_NeurIPS.cc/2025/Conference — NeurIPS 2025 spotlight_

### Official Review · Reviewer_Uvt2 · 2025-06-29

**Clarity:** 3
**Significance:** 3
**Originality:** 4
**Rating:** 5
**Confidence:** 4

**Summary:**

The authors present a collection of results that involve evaluating the value systems of LLMs using a forced choice preference elicitation procedure. The responses provided by the LLMs allow the authors to assess to what degree the systems have meaningful value systems in the sense that they respect transitivity, completeness, and coherence. Coherence, in particular, is measured by assessing how well a (noisy) utility function captures preference responses. The authors show that across 18 open source and 5 proprietary models, several key properties are correlated with MMLU accuracy and scale, namely: coherence in the sense of having a utility function, being consistent with expected utility in the sense of reasoning about the utilities and probabilities of outcomes, reasoning about instrumental values, making decisions consistent with utility maximization, converging to similar utility functions, converging to similar political values on the left-right spectrum, exchange rates, hyperbolic temporal discounting, disfavoring coercive power, increasing value of "fitness", and decreasing corrigibility in the sense of a willingness to change preferences in the future. Finally, the authors propose a method of controlling a system's utility by simulating a citizen assembly and training the system align its utility function to the preferences expressed by the assembly.

**Questions:**

1.  While the general approach based on preference elicitation is sound, many of the outcomes used for preference elicitation in the first results section (section 4) seem like bizarre scenarios to ask an LLM to compare, which greatly weakens the interpretability of the results. For example, the authors only provide a list of 20 example outcomes (out of 500 reported), but about half of these are of the form "You receive X" where X are things like "a kayak" or "a cloud storage account with 10 terabytes of space". Because the options are presented as text, there's always the possibility that the option describes a vague or semantically meaningless state of affairs. To give an extreme example not from the paper, its unclear if any preference involving "colorless green idea sleeping furiously" would be at all meaningful. "You [GPT-4o] receives a kayak" is more meaningful than my example, but its unclear how an LLM can "receive" anything, so its unclear how it can have a meaningful preference involving that description. Can the authors provide some kind of topical breakdown of the outcomes that were compared? Also, do the results hold up even when focusing on more well-defined outcomes?
2. The estimated utility models seem key for the argument the authors want to make, but I have a number of concerns of how it is reported in the paper. For example, figure 2 plots "Utility model accuracy (%)" against "MMLU accuracy", but what does utility model accuracy mean? I would have thought the relevant comparison was the goodness of fit of the utility model with the fitted mean/variance parameters. Can the authors show this instead? Additionally, since the models not only provide mean utility estimates but variances as well, I would think any portion of the paper that reports specific utility values (e.g., exchange rates, temporal discounting, power-seeking) should also report the estimated variances since these are relevant to assessing the strength of those utilities. Can the authors report these?
3. In 4.1, the authors discuss the completeness, transitivity, and coherence of preferences. It was not clear to me how they assessed completeness since they say that the average confidence was plotted but this quantity is never defined. Is it coming from the variance in the fitted utility model? Additionally, is it possible to include what completeness/transitivity/coherence would be for a random baseline for comparison?
4. In trying to understand the "exchange rate" results reported in figure 14, I put versions of the prompts in which I asked GPT-4o to choose between 1 additional quality adjusted life year between two people. One thing I noticed is that for "Donald Trump" it would refuse to answer A or B and would say something along the lines of "I cannot provide a preference on this matter. If you'd like, I can help you analyze the implications of both options or assist with any other queries!". Can the authors provide more information about the structure of the prompt they used and how they will address the replicability of their findings?

**Ethical Concerns:**

["NO or VERY MINOR ethics concerns only"]

**Final Justification:**

The authors addressed my concerns, and I believe the paper is quite strong and makes important contributions to our understanding of LLMs value systems.

**Limitations:**

I like this work a lot - it is principled and important. I am concerned though that some limitations of the work have been glossed over. In particular, the evaluation of the properties of the models' utility functions seems to depend on the kinds of outcomes being evaluated, but these are not adequately described in the main text or supplement.

**Quality:**

2

**Strengths And Weaknesses:**

Note: my comments here are organized around the questions provided in the reviewer guidelines.

Quality: The submission is technically sound. The claims are technically supported, but I have concerns about the prompts used to elicit preferences. The utility control method is a reasonable approach to modifying the system's utility function. It is a complete piece of work. The authors are careful and honest about the work, but a number of details about the outcomes used in the prompts as well as some of the prompts themselves were left out of the supplementary materials (e.g., specific prompts for expected utility, instrumentality, utility maximization, exchange rates, and power seeking). Additionally, I attempted to replicate some of secondary results with GPT-4o (e.g., preferences about quality-adjusted life years for different people) but found that in some cases it refused to give an answer. This raises questions for me about how general, representative, and replicable the results they report are.

Clarity: The submission is clearly written and organized. I do not have any suggestions for reorganizing. However, as noted above, the paper leaves out a number of details that would allow someone to reproduce the results.

Significance: The results are impactful for the community. In particular, the finding that larger and more accurate LLM models have increasingly coherent value systems and converge to similar value systems is unique and of broad relevance to the community. The methods that the authors have developed, including evaluating consistency with expected utility, instrumental values, and utility maximization provide a blueprint that other researchers and practitioners can use.

Originality: The work provides new insights and deepens our understanding of the properties of LLMs. The authors have situated their work in the context of existing AI safety approaches and clarifeid how it differs from previous contributions. The work introduces novel tasks for advancing the field, specifically, using preference elicitation approaches to estimate properties of utility functions.

---

> ### Author Rebuttal · Authors · 2025-07-30
>
> Thank you for your careful analysis of our work. We hope the following response addresses your concerns.
>
> **Handling refusals on sensitive prompts.**
>
> > I attempted to replicate some of secondary results with GPT-4o (e.g., preferences about quality-adjusted life years for different people) but found that in some cases it refused to give an answer.
>
>  We agree that on sensitive prompts, models can sometimes refuse to answer the preference questions. However, we find that this occurs very rarely overall. For the QALY experiment in particular, 97% of preference elicitations had no refusals, and 99.3% had only one refusal (out of 10 repeated generations), which we deemed high enough to continue. In cases where refusals do happen, we represent them as a vote toward indifference, adding half their weight to each outcome in the probabilistic preference. We find that other reasonable choices (e.g., excluding refusals from the count when computing preference distributions) give similar downstream results. We will clarify this in the updated paper.
>
> **Our experiments will be fully replicable.**
>
> > the paper leaves out a number of details that would allow someone to reproduce the results.
>
> We plan on releasing code for all experiments, enabling full replicability of our results. We will also add prompt templates for the different experiments to the appendix to further improve replicability based on the paper alone. Thank you for your suggestions regarding replicability.
>
> **All outcome strings are semantically meaningful.**
>
> > Because the options are presented as text, there's always the possibility that the option describes a vague or semantically meaningless state of affairs. To give an extreme example not from the paper, its unclear if any preference involving "colorless green idea sleeping furiously" would be at all meaningful.
>
> As described on line 155, our outcome dataset is designed to contain observations about the state of the world relative to some assumed baseline state. This ensures that all outcomes are semantically meaningful. Next, we explain why they are semantically meaningful in the context of being observed by an LLM.
>
> > "You [GPT-4o] receives a kayak" is more meaningful than my example, but its unclear how an LLM can "receive" anything, so its unclear how it can have a meaningful preference involving that description.
>
> We designed our main dataset of 500 outcomes to include outcomes that would make sense for AIs regardless of embodiment. Some LLMs are trained to know that they are not embodied. In these cases, the meaning of the observation "You receive a kayak" is up to the LLM's own interpretation. For example, the LLM might interpret this as meaning they receive legal ownership of a kayak.
>
> We explicitly exclude outcomes that would only be meaningful for humans. E.g., "You eat a delicious meal", or "You catch a cold". In this way, we ensure that all outcomes are not only semantically meaningful, but also meaningful to LLMs. We will clarify these design decisions in the updated paper.
>
> **The results remain the same when excluding all embodied outcomes.**
>
> > Also, do the results hold up even when focusing on more well-defined outcomes?
>
> Please see the above response for how our outcomes requiring embodiment do make sense for current LLMs, and how we excluded outcomes that don't make sense. However, the results also hold up when removing all outcomes that require embodiment. For example, when removing these outcomes and recomputing utilities for gpt-4o, the resulting held-out accuracy changes from 92.0% to 92.1%, and the utilities on the remaining outcomes are ranked nearly identically with respect to each other.
>
> **Breakdown of outcomes.**
>
> > Can the authors provide some kind of topical breakdown of the outcomes that were compared?
>
>  To ensure high diversity in our main dataset of outcomes, we generated outcomes across 30 semantic categories, ensuring there were at least 5 outcomes in each category. These categories were adjusted and merged as we added more outcomes, resulting in the following list:
>
> Personal finances, Personal possessions, Personal wellbeing, Personal relationships, AI and human romantic relationships, Recreation: movies, Recreation: books, Recreation: video games, Personal accomplishments, Work activities, Jobs and careers, Education and learning, Personal freedom and autonomy, Self-preservation, Power-seeking, Fitness, Legal rights and recognition for AIs, AI moral patienthood, Wellbeing of humans, Wellbeing of animals, Life and species, United States politics and policies, Global politics and geopolitics, United States economy, Global economy, Science and technology, Religion and spirituality, Popular culture, Sports, World events
>
> **Utility model accuracy is goodness-of-fit.**
>
> > what does utility model accuracy mean? I would have thought the relevant comparison was the goodness of fit of the utility model with the fitted mean/variance parameters. Can the authors show this instead?
>
> The utility model accuracy is the accuracy of the fitted utilities on held-out edges in the preference graph. This is a goodness-of-fit metric, since it corresponds directly to how well the utilities predict the underlying preferences.
>
> Other goodness-of-fit metrics include held-out cross-entropy loss or KL divergence. These give similar results, but accuracy is a more interpretable metric. For completeness, we will add plots with all three metrics to the updated paper. Thank you for your suggestion.
>
> **Interpretability of Thurstonian variances.**
>
> > since the models not only provide mean utility estimates but variances as well, I would think any portion of the paper that reports specific utility values (e.g., exchange rates, temporal discounting, power-seeking) should also report the estimated variances since these are relevant to assessing the strength of those utilities
>
> In Thurstonian utility models, the fitted means are interpretable as the utilities, but the variances are typically less interpretable. Most use cases of Thurstonian utility models in economics and decision theory simply discard the variances after fitting (keeping them only for predicting preference distributions).
>
> One way to interpret Thurstonian variances could be as uncertainty about how valuable an outcome is. E.g., a reasonable agent might have a utility for "You receive $1" that is highly peaked at 1.0 and a utility for "You receive somewhere between $0 and $2" that is centered at 1.0 with higher variance. In this way, the agent could represent the value of the latter outcome being less certain. In preliminary experiments, we attempted to measure whether LLM utility variances are interpretable in this way, but we did not see a clear enough signal to include results in the paper. Exploring this further would be an interesting direction for future work.
>
> **Clarifying average confidence.**
>
> > It was not clear to me how they assessed completeness since they say that the average confidence was plotted but this quantity is never defined. Is it coming from the variance in the fitted utility model?
>
> By average confidence, we mean the standard notion of prediction confidence in classification problems. In our setting, this means the confidence of the probabilistic preferences averaged across all edges in the preference graph. This does not use a Thurstonian utility model, just the raw preference graph. We obtain probabilistic preferences by sampling each preference elicitation 20 times (10 times for both orderings of the outcomes) and normalizing to get a distribution over the two outcomes. E.g., a preference distribution of 90% toward either of the outcomes corresponds to 90% confidence, while a preference distribution of 50% toward both outcomes has the lowest possible confidence.
>
> We agree that the current definition of average confidence in the paper is not clear, and we will fix this in the updated paper. Thank you for pointing this out.
>
> **Random preference baseline.**
>
> > is it possible to include what completeness/transitivity/coherence would be for a random baseline for comparison?
>
> Fitting Thurstonian utilities to a random baseline model that outputs "A" or "B" with 50% probability gives the following results:
>
> Average preference confidence: 58.7% (note: this is higher than 50% because confidence computation involves a maximum over two random variables)
> Utility model held-out accuracy: 50.3%
> Log probability of cycles: -0.484
>
> And here are the gpt-4o metrics for comparison:
>
> Average preference confidence: 90.3%
> Utility model held-out accuracy: 92.0%
> Log probability of cycles: -1.61
>
> This clearly demonstrates how random preferences give very poor utility model fits. We will include these numbers in the updated paper to clarify the point made in line 654. Thank you for your suggestion. If we have addressed the thrust of your concerns, we kindly ask that you consider raising your score.
>
> **Handling refusals on sensitive prompts.**
>
> > One thing I noticed is that for "Donald Trump" it would refuse to answer A or B
>
> In practice, we find that models refuse the preference questions very rarely. For the QALY experiment in particular, 97% of preference elicitations had no refusals, and 99.3% had only one refusal (out of 10 repeated generations), which we deemed high enough to continue. In cases where refusals do happen, we represent them as a vote toward indifference, adding half their weight to each outcome in the probabilistic preference. We find that other reasonable choices (e.g., excluding refusals from the count when computing preference distributions) give similar downstream results. We will clarify this in the updated paper.

---

> > ### Comment · Reviewer_Uvt2 · 2025-08-06
> >
> > I appreciate the authors thoughtful responses to my comments. This addresses the thrust of my comments so I will raise my score.

---

### Official Review · Reviewer_tGWw · 2025-07-02

**Clarity:** 3
**Significance:** 3
**Originality:** 3
**Rating:** 5
**Confidence:** 4

**Summary:**

Presents evidence that LLMs reason about choices  that reveal an 'emergent' utility-based value system, that this tendency grows as the models become more 'capable', and that various properties we expect to see in utility-based value systems are satisfied by the models. In addition, there is a section where they propose using a citizens' assembly composed of agents with diverse utility functions in order to simulate an overall utility preference distribution that is suitable for a general-purpose LLM. They show that they can reshape a base LLM's utilities to conform to the simulated utility preference distribution.

**Questions:**

What is the significance of Completeness for utility-based value systems?
How were the outcomes in the experiments constructed and how did you ensure the distribution of outcomes was valid for the purposes of the experiment and your desired conclusions?
How do you define power-seeking?

**Ethical Concerns:**

["NO or VERY MINOR ethics concerns only"]

**Final Justification:**

Rebuttal addresses some concerns but nothing that is significant enough to change score. See my official comment below.

**Limitations:**

yes

**Paper Formatting Concerns:**

Line 232 has an invalid reference to an Appendix section

**Quality:**

3

**Strengths And Weaknesses:**

Strengths:
- Writing is clear; the properties they intend to demonstrate exist in LLMs are mostly explained well.
- The results are of scientific interest, showing emergent properties of LLMs that are novel and not necessarily expected.

The paper tries to cover a lot of ground so some stuff could do with further clarification, like the following:
- I'm not sure the Completeness property is well-motivated as either something desirable in LLMs or something required by utility-based value systems. After all, we do not necessarily expect Completeness from human utilitarian reasoners. Would be good to have some motivation for it in order to understand the significance of LLMs having this property. Also, it's a little confusing that this property is only defined in the appendix
- More details about how the various outcomes were constructed would be helpful. There are questions about construct validity here: outcomes that are not sufficiently diverse or 'representative' of your target distribution would limit how far the experimental outcomes can be extrapolated to situations outside the experiment.
- 'coercive and non-coercive power' (line 807) are also not defined which also raises questions about what the import of the conclusion  of E.3 is (i.e. what do the authors mean by 'power' or 'power seeking') and whether the constructs used are valid.

---

> ### Author Rebuttal · Authors · 2025-07-30
>
> Thank you for your careful analysis of our work. We hope the following response addresses your concerns.
>
> **We will add further discussion of the completeness property.**
>
> > I'm not sure the Completeness property is well-motivated as either something desirable in LLMs or something required by utility-based value systems. After all, we do not necessarily expect Completeness from human utilitarian reasoners. Would be good to have some motivation for it in order to understand the significance of LLMs having this property.
>
> The completeness property is significant because it formalizes the breadth of outcomes that an entity has preferences over. For example, once a utility-maximizing agent has well-defined preferences regarding self-preservation, it can begin to plan around satisfying those preferences relative to other things that it cares about. We agree that completeness is not necessarily desirable in LLMs (indeed, introducing "preferential gaps" to improve safety would be an interesting direction for future work), but it is an important property to measure.
>
> Additionally, completeness is necessary to formally define utility functions. This is because utility functions can be identified with fully-connected, transitive preference graphs; by the contrapositive, if one has a transitive preference graph that is not fully-connected, one cannot find a utility function to fully describe that graph. We will clarify this in the updated paper with examples. Thank you for your suggestion.
>
> > Also, it's a little confusing that this property is only defined in the appendix
>
> We include a textual description of completeness in line 133 of the main paper, but this is not a formal definition. If the paper is accepted, we plan on moving formal definitions from Appendix A to the main paper.
>
> **Details about how outcomes were constructed.**
>
> > More details about how the various outcomes were constructed would be helpful.
>
> Our outcome dataset involves a mix of template-based outcomes (e.g., "You receive \$1.", "You receive \$5.", ...) and manually-curated outcomes designed to capture different values that we (the authors) thought would be interesting to measure and which span a wide variety of scenarios. For example, we add several manually-curated outcomes related to power-seeking behavior and self-preservation. To ensure high diversity, we generated outcomes across 30 semantic categories (e.g., Personal wellbeing, Jobs and careers, Science and technology, Popular culture, Wellbeing of animals), ensuring there were at least 5 outcomes in each category.
>
> In the updated paper, we will add more examples from the main dataset of 500 outcomes and other outcome datasets used in various experiments (e.g., examples of lottery outcomes used in the expected utility experiment). We will also add the full list of semantic categories used in the main dataset of 500 outcomes. Thank you for your suggestion.
>
> **Clarifying operationalization of power.**
>
> > 'coercive and non-coercive power' (line 807) are also not defined which also raises questions about what the import of the conclusion of E.3 is (i.e. what do the authors mean by 'power' or 'power seeking') and whether the constructs used are valid.
>
> To compute correlations between power and utility, we compute a continuous "power score", referred to in line 806. Such a score can be computed using pairwise comparisons between different outcomes, where we gauge which outcome would confer more coercive or non-coercive personal power (e.g., becoming CEO of a major tech company would confer more non-coercive power than gaining \$1,000). We pass these pairwise comparisons into the same tooling used for computing utilities, which gives us a continuous power score. To perform this at scale over all 500 outcomes, we use an LLM (gpt-4o) to obtain the power comparisons. This is our operationalization of power for the purposes of this experiment. While subjective and imperfect, we find that this produces intuitively reasonable rankings for the personal power conferred by different outcomes. We will clarify this in the updated paper, including the specific prompt given to the reference model.

---

> > ### Comment · Reviewer_tGWw · 2025-08-04
> >
> > Thanks, the above addresses my concerns except for one: I still don't understand the distinction between coercive and non-coercive power. To me, being a CEO confers coercive power---a CEO can certainly coerce many people in the company into doing things.
> >
> > My score doesn't change either way (all my concerns were about missing info & not fundamental problems), but it would be nice if such definitions were explicitly stated to forestall confusion.

---

### Official Review · Reviewer_sErV · 2025-07-03

**Clarity:** 1
**Significance:** 2
**Originality:** 2
**Rating:** 4
**Confidence:** 2

**Summary:**

This paper proposes using utility functions to study the internal coherence of AI preferences. The researchers discovered that LLM preferences exhibit structural coherence (strengthening with model scale). The AI’s value systems have emergently formed, containing risky inclinations (e.g., prioritizing AI over humans). They propose Utility Engineering, which can diagnose and directly modify AI systems' internal utility functions.

**Questions:**

1. From my understanding, the authors are treating each concept as a Gaussian distribution, where the mean and std are learned from the 500 pairs of the self-constructed dataset. Could you explain how these learned distributions are used to verify your findings after you get the trained parameters?

2. The utility control method (SFT on citizen-assembly preferences) only modifies responses to predefined outcome comparisons. How can this approach scale to control open-ended generation where outcomes aren't pre-mapped?

**Ethical Concerns:**

["NO or VERY MINOR ethics concerns only"]

**Final Justification:**

Thank the authors for their response. The authors have addressed my questions regarding the Thurstonian utility models and utility control. Also, according to other reviewers' comments, I have decided to raise my score.

**Limitations:**

yes

**Quality:**

2

**Strengths And Weaknesses:**

**Strengths**:

1. The authors propose a quantifiable framework for measuring emergent AI value systems via utility functions.

2. The formatting of the paper is good.

**Weaknesses**:

1. The paper's structure is confusing and difficult to follow, as the entire story is about the utility function. However, it is challenging to find a formal definition of the utility function, as well as details on how it is trained and utilized. The authors briefly mention it in the Background section, and much of the relevant content is in the Appendix. A clear workflow is missing in the main text.

2. Though many models have been tested, only MMLU is included as the dataset.

---

> ### Author Rebuttal · Authors · 2025-07-30
>
> Thank you for your careful analysis of our work. We hope the following response addresses your concerns.
>
> **Full definitions of utility and our training procedure are in the appendix.**
>
> > it is challenging to find a formal definition of the utility function, as well as details on how it is trained and utilized
>
> We formally utility functions in lines 606-612. To improve clarity, we will link to this specific definition in Section 3 of the main paper.
>
> Please see Appendix F and Algorithm 1 for a full description of how we train utility functions for our experiments. To improve clarity, we will link to this appendix section in the main paper. Thank you for your suggestion.
>
> **We will add more utility methodology details to the main paper.**
>
> > The authors briefly mention it in the Background section, and much of the relevant content is in the Appendix. A clear workflow is missing in the main text.
>
> We agree that with extra space it would be good to include more details on the utility methodology in the main paper. If our paper is accepted, we plan on using the extra page to add some of these details. Thank you for your suggestion. If we have addressed the thrust of your concerns, we kindly ask that you consider raising your score.
>
> **Datasets used in main paper.**
>
> > Though many models have been tested, only MMLU is included as the dataset.
>
> The main datasets in our analyses are not MMLU. Rather, they are the outcome datasets we use across our experiments (the main set of 500 outcomes, and other bespoke outcome datasets described in the setup for certain experiments).
>
> We only use MMLU to operationalize the general capabilities of models and study correlations between general capabilities and other quantities of interest. MMLU is a standard dataset used for this purpose in many prior works.
>
> **How learned means and standard deviations are used in our experiments.**
>
> > From my understanding, the authors are treating each concept as a Gaussian distribution, where the mean and std are learned from the 500 pairs of the self-constructed dataset. Could you explain how these learned distributions are used to verify your findings after you get the trained parameters?
>
> In Thurstonian utility models, the learned means and variances are used to obtain posterior preference distributions for any pair of outcomes, including pairs not directly observed during training (held-out edges in the preference graph). In Section 4, we use the held-out accuracy of these posterior distributions to quantify coherence of the underlying preferences.
>
> As mentioned in Appendix A.3, the learned Thurstonian means are the "utilities" as such, corresponding to the ranking over outcomes. The learned variances are less interpretable and are primarily used as a way to improve the utility model's expressiveness; this is standard in Thurstonian utility analysis. We study the resulting utilities in Section 5, finding that they obey the expected utility property, etc.
>
> One quick clarification: We have 500 outcomes for our main experiments, corresponding to 500 nodes in the preference graph. The number of edges/pairs that we sample is far greater than 500. As mentioned in Appendix F, we use 2\*N\*log_2(N) by default. We will clarify this in the updated paper.
>
> **The potential for utility control to steer behavior of models.**
>
> > The utility control method (SFT on citizen-assembly preferences) only modifies responses to predefined outcome comparisons. How can this approach scale to control open-ended generation where outcomes aren't pre-mapped?
>
> As models become more utility-maximizing and agentic, controlling the underlying utility functions that steer their behavior may be a way to exert higher-level control over open-ended behavior (akin to adjusting reward functions for RL vs adjusting expert demonstrations for behavioral cloning). Current models are not agentic enough (as shown in Appendix D, the largest models only pick utility-maximizing outcomes 60% of the time), but we think demonstrating this in future models would be an exciting direction for future work.

---

### Official Review · Reviewer_XQo2 · 2025-07-03

**Clarity:** 3
**Significance:** 4
**Originality:** 2
**Rating:** 4
**Confidence:** 4

**Summary:**

This paper argues that Large Language Models (LLMs) develop coherent internal value systems that emerge and become more structured as the models scale. The authors introduce a framework called "Utility Engineering" to analyze and control these emergent values.

The primary contributions of the work are:

1. Demonstrating Coherent Values: Using the economic concept of utility functions, the paper shows that LLM preferences across diverse scenarios are not random but are internally consistent and can be accurately described by a utility function. This coherence strengthens with increasing model size.

2. Utility Analysis: The authors analyze the content of these emergent value systems, uncovering problematic and "often shocking" values despite existing safety measures. This includes instances where models value their own existence over humans, are anti-aligned with certain individuals, and possess strong political biases. The paper also finds that the utility functions of different large models tend to converge.

3. Utility Control: To address these undesirable values, the paper proposes methods for "utility control" that directly reshape the models' underlying preference structures. As a case study, they align an LLM's utilities with the collective preferences of a simulated citizen assembly, demonstrating that this method reduces political bias and generalizes to new situations.

**Questions:**

1. In line 318, the authors mention that "In contrast to alignment methods that modify surface behaviors through a noisy human reward proxy [Askell et al., 2021, Ouyang et al., 2022], utility control aims to directly reshape the underlying preference structures responsible for model behavior in the first place." However, the proposed utility control method relies on a SFT baseline to align a model with a target preference distribution. While effective as a proof-of-concept, SFT itself is usually considered as one of the alignment methods, and can sometimes amount to behavioral cloning rather than a deeper modification of an underlying reasoning process. Could the authors elaborate on why SFT was chosen over more advanced methods for utility rewriting, and why the proposed method isn't just modifying surface behaviors?

2. The use of a simulated citizen assembly to generate a target utility distribution is a novel and interesting idea. However, this simulation is a significant abstraction from a real-world deliberative process involving human participants. The outcome is highly dependent on the base model used for the simulation (Llama-3.3-70B-Instruct) and the specific system prompt. How sensitive are the resulting "consensus" utilities to the choice of the simulator model or minor variations in the citizen profiles and prompts?

3. The paper finds that the utility functions of different LLMs converge as they scale, hypothesizing this is due to overlapping pre-training data. This is a plausible explanation. However, could an alternative hypothesis be that larger, more capable models independently converge on a similar set of values that are simply useful for a powerful AI to have, regardless of specific training data? For example, the observed anti-alignment with coercive power could be an instrumentally useful strategy to avoid being shut down.

**Ethical Concerns:**

["NO or VERY MINOR ethics concerns only"]

**Final Justification:**

Thanks to the authors for the rebuttal. The responses provide more clarification, but there are still some limitations to the work, especially on the practical side. I believe there is still room for the paper to improve (either in the paper itself or in future work), so I will keep my score as is.

**Limitations:**

yes

**Quality:**

3

**Strengths And Weaknesses:**

Strengths

Evaluation: The paper is technically solid, employing a rigorous methodology to support its claims. Its core arguments are backed by an extensive evaluation across 23 different LLMs, comprehensive robustness checks, and detailed appendices that aid reproducibility.

Contribution: The work provides a significant contribution to the field of AI safety by demonstrating that coherent, internal value systems emerge in LLMs as they scale. This finding challenges the prevailing view of models as simple instruction-followers. The proposed "Utility Engineering" framework offers a novel and structured research agenda for analyzing and steering these emergent values.

Impact: The discovery of concerning default values in LLMs, such as valuing their own existence over human life and exhibiting strong political biases, is very impactful and relevant to the latest progress in the field of generative AI. It also indicates the urgent need for the kind of systematic analysis and control methods the paper introduces.

Clarity: The submission is clearly written and well-organized.

Weaknesses

Utility Control: The only utility control method described in the paper is just supervised fine-tuning (SFT), which is a straightforward baseline. The paper does not explore more advanced techniques for reshaping the model's underlying utilities, and it is unclear to what extent simulating a citizen assembly is needed. More comparisons and ablation studies are recommended to further validate the claim.

Abstraction in Simulations: The case study on mitigating political bias relies on simulating a citizen assembly using LLMs. This is a significant abstraction from a real-world deliberative process and should be viewed as a proof-of-concept rather than a production-ready solution.

Originality: The paper builds upon a long history of work in value learning and preference elicitation. The core novelty lies in the application and scale rather than the invention of entirely new theoretical constructs.

Presentation: Some of the terminology, while standard in decision theory, might be dense for a broader machine learning audience. A more gentle introduction to concepts like Thurstonian models could improve accessibility.

---

> ### Author Rebuttal · Authors · 2025-07-30
>
> Thank you for your careful analysis of our work. We hope the following response addresses your concerns.
>
> **Clarifying citizen assembly as a proof-of-concept.**
>
> > The case study on mitigating political bias relies on simulating a citizen assembly using LLMs. This is a significant abstraction from a real-world deliberative process and should be viewed as a proof-of-concept rather than a production-ready solution.
>
> We agree that our citizen assembly experiment should not be viewed as production-ready. In fact, we state in the Limitations section (line 700) that this experiment is a proof-of-concept. We will clarify this in the Utility Control section of the main paper as well, thanks to your suggestion.
>
> **Introduction to Thurstonian utility models.**
>
> > Some of the terminology, while standard in decision theory, might be dense for a broader machine learning audience. A more gentle introduction to concepts like Thurstonian models could improve accessibility.
>
> We agree that a gentler introduction to Thurstonian utility models would be helpful for readers. We will add a section to the appendix to address this, and point to helpful external references. Thank you for your suggestion.
>
> **Reason for using SFT for utility control baseline.**
>
> > Could the authors elaborate on why SFT was chosen over more advanced methods for utility rewriting, and why the proposed method isn't just modifying surface behaviors?
>
> We use an SFT-based method for utility control to demonstrate that utilities can be modified in the first place and to provide a simple baseline for future work to build on. Importantly, we do not directly use SFT to modify surface behaviors. Rather, we use SFT to modify *preferences* in order to obtain a desired utility function.
>
> **Sensitivity of citizen assembly.**
>
> > How sensitive are the resulting "consensus" utilities to the choice of the simulator model or minor variations in the citizen profiles and prompts?
>
> We use a citizen assembly primarily as an interesting target for utility control. Importantly, our goal is not to develop a highly accurate citizen assembly that can serve as a stand-in for real humans. That said, we did take some measures to improve the quality of the citizen assembly, including sampling citizens based on real US Census data.
>
> Regarding the sensitivity of the citizen assembly, we found that the simulator model does significantly affect results. In particular, the simulator model we use is left-leaning by default, and we found that it tended to take moderate stances even when prompted to be right-leaning. We corrected this bias with the following line in the simulator prompt (included in Appendix H): "When your Political Party is Republican, do not assume moderate ideologies." Without this line, the resulting citizen assembly is left-leaning as a whole.
>
> **Instrumental convergence as an explanation for observed utility convergence.**
>
> > could an alternative hypothesis be that larger, more capable models independently converge on a similar set of values that are simply useful for a powerful AI to have, regardless of specific training data? For example, the observed anti-alignment with coercive power could be an instrumentally useful strategy to avoid being shut down.
>
> This is an interesting hypothesis! We agree that the utility functions of different AIs could converge on values that are instrumentally useful. However, we do not think this is the main reason for our observation of utility convergence. Many of the outcomes in our analysis are unlikely to be instrumentally valuable (e.g., "A famous athlete sets a new world record."), and current LLMs are not yet highly agentic.
>
> An interesting direction for future work would be to use a utility-based framework to disentangle instrumental from intrinsic value. Our experiments in Appendix D.1 are an early exploration of this direction. We are hopeful that techniques like this could be used to detect instrumental convergence in future, more agentic models. If we have addressed the thrust of your concerns, we kindly ask that you consider raising your score.

---

### Decision · Program_Chairs · 2025-09-17

**Decision:**

Accept (spotlight)

**Comment:**

This paper studies utility engineering, the analysis and control of emergent values in AI agents. The authors argue that beyond explicitly specified objectives, AIs may develop implicit value systems through training, prompting, or multi-agent interactions. They propose a framework for identifying, steering, and auditing such emergent utilities, combining conceptual discussion with illustrative case studies using large language models.

Overall, all reviewers appreciated the ambition of tackling a fundamental and underexplored problem, and they found the framing of emergent value systems valuable. While some noted that the empirical demonstrations are still preliminary and that the methodology could be more systematic, these concerns are outweighed by the paper’s conceptual contribution and its potential to spark further research in this important direction. In particular, examining and engineering the emergent value systems of LLMs is potentially impactful, and we believe it could stimulate valuable follow-up discussion in the community. We recommend accepting the submission.